# *Lysimachia mauritiana* Lam. Extract Alleviates Airway Inflammation Induced by Particulate Matter Plus Diesel Exhaust Particles in Mice

**DOI:** 10.3390/nu16213732

**Published:** 2024-10-31

**Authors:** Yoon-Young Sung, Seung-Hyung Kim, Won-Kyung Yang, Heung Joo Yuk, Mi-Sun Kim, Dong-Seon Kim

**Affiliations:** 1KM Science Research Division, Korea Institute of Oriental Medicine, 1672 Yuseongdae-ro, Yuseong-gu, Daejeon 34054, Republic of Korea; yysung@kiom.re.kr (Y.-Y.S.); yukhj@kiom.re.kr (H.J.Y.); misun210@gmail.com (M.-S.K.); 2Institute of Traditional Medicine and Bioscience, Daejeon University, 62 Daehak-ro, Dong-gu, Daejeon 34520, Republic of Korea; sksh518@dju.kr (S.-H.K.); ywks1220@dju.kr (W.-K.Y.)

**Keywords:** airway inflammation, lung, neutrophil, PM10D, respiratory disease

## Abstract

Exposure to air pollution poses a risk to human respiratory health, and a preventive and therapeutic remedy against fine dust-induced respiratory disease is needed. Background/Objectives: The respiratory-protective effects of *Lysimachia mauritiana* (LM) against airway inflammation were evaluated in a mouse model exposed to a fine dust mixture of diesel exhaust particles and particulate matter with a diameter of less than 10 µm (PM10D). Methods: To induce airway inflammation, PM10D was intranasally injected into BALB/c mice three times a day for 12 days, and LM extracts were given orally once per day. The immune cell subtypes, histopathology, and expression of inflammatory mediators were analyzed from the bronchoalveolar lavage fluid (BALF) and lungs. Results: LM alleviated the accumulation of neutrophils and the number of inflammatory cells in the lungs and the BALF of the PM10D-exposed mice. LM also reduced the release of inflammatory mediators (MIP-2, IL-17, IL-1α, CXCL1, TNF-α, MUC5AC, and TRP receptor channels) in the BALF and lungs. Lung histopathology was used to examine airway inflammation and the accumulation of collagen fibers and inflammatory cells after PM10D exposure and showed that LM administration improved this inflammation. Furthermore, LM extract inhibited the MAPK and NF-κB signaling pathway in the lungs and improved expectoration activity through an increase in phenol red release from the trachea. Conclusions: LM alleviated PM10D-exposed neutrophilic airway inflammation by suppressing MAPK/NF-κB activation. This study indicates that LM extract may be an effective therapeutic agent against inflammatory respiratory diseases.

## 1. Introduction

The presence of fine particulate matter (PM), a type of air pollution, poses a danger to human health by damaging the respiratory system, including the airway and lungs [1]. PM with a diameter of 2.5–10 μm includes diverse chemical constituents, such as sulfates, nitrates, ammonium, organic and elemental carbon, biological compounds, organic compounds (e.g., polycyclic aromatic hydrocarbons), and metals (e.g., copper, nickel, vanadium, zinc, and cadmium) [2]. Diesel exhaust particles (DEPs), consisting of a combination of elemental carbon, organic compounds (including polycyclic aromatic hydrocarbons), sulfates, nitrates, and metals, are the main constituent of PM with a diameter of less than 2.5 μm (PM2.5) and are associated with airway inflammation and remodeling as well as heart dysfunction [3]. Because this fine PM can cause chronic respiratory diseases, it is crucial to investigate ways to prevent and treat respiratory disorders caused by air pollution [4].

*Lysimachia mauritiana* (LM) is a dicotyledonous plant found in coastal rocky areas in the southern region of South Korea, including Jeju Island, Dokdo Island, and Ulleung Island [5,6]. LM-containing *Lysimachia* plants, known as seagrass, pearls vegetables, and beach pearls grass, live in relatively dry conditions, such as rocky crevices along the coast, and are considered to be medical plants due to their antioxidant, anticancer, antibacterial, and antiviral activities from the large amounts of flavonol glycosides (quercetin, kaempferol, and myricetin types) they contain [7,8,9,10]. Flavonol glycosides of hyperin and kaempferol-3-O-rhamnosyl-galactosides have been isolated from the whole plant of LM [7]. The young leaves and shoots of LM are eaten as vegetables, and the plant is also used for ornamental purposes. In oriental herbal medicine, LM is mainly used as a diuretic and for urinary diseases, menstrual irregularities, and labor pains. It is also known to be effective for trauma, high blood pressure, diabetes, constipation, swelling, bruising, and sore throat [11,12]. However, although LM has been used for respiratory diseases, the effect of LM on airway inflammation has not been examined. Thus, this study investigated the possible effects of LM extract on mice with airway inflammation induced by exposure to a fine dust mixture: a mixture of PM with a diameter of less than 10 μm (PM10) and DEP (PM10D).

## 2. Materials and Methods

### 2.1. Preparation of Lysimachia mauritiana and UPLC-QTof MS Analysis

LM extract was provided by Dongkook Pharmaceutical (Seoul, Republic of Korea). The LM extract was extracted from air-dried aerial parts (leaves) with 50% ethanol for 5-h. The extract was filtered, concentrated under reduced pressure, and then after mixing the food-grade maltodextrin, the mixture was spray-dried to obtain a powdered sample (code name DKB-139, batch No. 240425-001, S&D Co., Ltd, Cheongju-si, Chungcheongbuk-do, Republic of Korea). The 50% ethanol extract from *L. mauritiana* was analyzed using an AQUITYTM UPLC system (Waters Corp., Milford, MA, USA) equipped with a binary gradient system, an auto-injector, and a UV-Visible detector. The sample (2.0 μL) was separated on a BEH C18 column (2.1 × 100 mm, 1.7 μm) at a flow rate of 0.4 mL/min and eluted using a linear gradient of two mobile phases containing 0.1% formic acid (A: water; B: acetonitrile). A chromatographic gradient was optimized as follows: 0 min, 10% B; 0–8 min, 10–30% B; 8–11 min, 30–90% B; 11–12 min, 90–100% B; 12–13.3 min, 100% B; and at 13.4 min, it returned to 10% B and was maintained until the 15 min mark. Mass spectrometry analysis was conducted using a quadrupole time-of-flight mass spectrometer (Xevo G2 QToF, Waters Corp., Milford, MA, USA) with an electrospray ionization (ESI) interface operating in negative ion mode. The system was configured with the following settings: a cone voltage of 40 V, a capillary voltage of 2500 V, a source temperature at 110 °C, and a desolvation temperature at 350 °C. Leucine-enkephalin ([M−H]− *m*/*z* 554.2615) was used as the lock mass with a sprayer reference solution. All solvents used for extraction and chromatography were of a LC/GC-MS grade (J. T. Baker, Phillipsburg, NJ, USA).

### 2.2. Animal Experiments

Experiment 1: Seven-week-old male BALB/c mice (Orient Bio, Seongnam, Republic of Korea) were maintained in the specific pathogen-free facility at 60% ± 10% humidity and 21 °C ± 2 °C. The experiment was performed according to the Guide for the Care and Use of Laboratory Animals and approved by the Committee for Animal Welfare at Daejeon University (DJUARB2022-041). Respiratory damage was induced by intranasal injection of PM10D (MilliporeSigma, Burlington, MA, USA) in aluminum hydroxide (1%) gel adjuvant, which included PM10 (ERMCZ120; 3 mg/mL) and DEP (NIST2975; 0.6 mg/mL), on days 4, 7, and 10 [13]. The mice were divided into five groups (*n* = 6/group): normal, PM10D control, PM10D and 3 mg dexamethasone/kg, PM10D and 50 mg LM extract/kg, and PM10D and 100 mg LM extract/kg [13]. The doses were determined from preliminary dose-dependent experiments. Dexamethasone (as a positive control) or LM extract was orally administered every day for 12 days. After completing the experiment, blood, bronchoalveolar lavage fluid (BALF), and tissues were isolated from the mice of each group under euthanasia. Figure 1A shows the schedule for determining the preventive effects of LM using an animal model of respiratory damage.

Experiment 2: The study of phenol red secretion for expectorant activity was carried out as previously reported [14]. The experiment was approved by the Animal Ethics Committee for the Korea Institute of Oriental Medicine (approval code 23-054). Seven-week-old male ICR mice (Orients Bio, Seongnam-Si, Republic of Korea) were divided into five groups (*n* = 12): normal, phenol red control, phenol red and 200 mg Levosol/kg (PharmGen Science, Seoul, Republic of Korea), phenol red and 50 mg LM extract/kg, and phenol red and 100 mg LM extract/kg. Levosol or LM extract was administered orally once a day for 3 days. The vehicle was administered to normal and control mice. Except for the normal group, phenol red (5%) was injected intraperitoneally in each mouse on day 3. The trachea was removed after 60 min and incubated in saline (1 mL) for 1 h to extract phenol red and then treated with sodium hydroxide (0.1 mL) to stop the reaction. The amount of secreted phenol red was quantified by measuring the absorbance at 546 nm.

### 2.3. Collection of Cells from BALF and Lung

The BALF was collected by trachea cannulation and aspiration. The differential cell count was examined in a cytospin with Diff-Quick staining. The lungs were treated with collagenase IV (1 mg/mL), then filtered and centrifuged, and the lung cells were collected [4].

### 2.4. Flow Cytometry Analysis

The lung cells or BALF cells were incubated with individual fluorescent-conjugated antibodies against lymphocytes, T cells, and B cells, and the cells were analyzed on a FACSCalibur (BD, Seoul, Republic of Korea) as performed in a previous study [14].

### 2.5. Release of Cytokines

The secretion levels of IL-17, IL-1α, macrophage inflammatory protein-2 (MIP-2), CXCL-1, and TNF-α in the BALF were measured using an enzyme-linked immunosorbent assay (R&D Systems, Minneapolis, MN, USA), and the absorbance at 450 nm was determined (Molecular Devices, San Jose, CA, USA).

### 2.6. Histopathological Analysis

The removed lungs were fixed, paraffinized, sectioned to a 5 μm thickness, and stained with Masson’s trichrome solution to examine collagen fiber formation or hematoxylin and eosin (H and E) solution to examine inflammation. The inflammatory severity was determined in a double-blind manner and rated on a subjective scale of 0–2 [13].

### 2.7. Quantitative Reverse Transcription-Polymerase Chain Reaction

Total RNA was isolated from the removed lungs using a HiGene precipitation assay (BIOFACT, Daejeon, Republic of Korea). The gene expression was quantified by quantitative reverse transcription-polymerase chain reaction with an SYBR Green Master Mix and primers using an Applied Biosystems 7500 Fast (Thermo Fisher Scientific, Waltham, MA, USA). The sequence of the primers is shown in Table 1 [13]. The transcript was expressed as ΔΔCt, normalized to β-actin.

### 2.8. Western Blot

Proteins from the lungs were extracted using PRO-PREP solution (Intron, Seongnam, Republic of Korea), and gel electrophoresis and membrane transfer were performed using a Bio-Rad Trans-Blot transfer system (Hercules, CA, USA). The membrane was blocked using a blocking buffer (EzBlock Chemi, ATTO, Daejeon, Republic of Korea) and then treated with phospho-NF-κB, -JNK, -p38, or -ERK and total-NF-κB, -ERK, -p38, or -JNK antibodies (Cell Signaling Technology, Danvers, MA, USA, 1:1000) for 24 h. The band was detected with chemiluminescence (Thermo Fisher Scientific, Waltham, MA, USA) and evaluated by ImageJ 7 software. The protein expression was expressed relative to β-actin.

### 2.9. Statistical Analysis

The data are expressed as the mean ± standard error of the mean. Statistical testing was performed by one-way analysis of variance and Duncan’s multiple comparison test using GraphPad Prism 7.0 software. *p*-values less than 0.05 were considered to demonstrate significant differences.

## 3. Results

### 3.1. Effects of LM Extract on Neutrophil Accumulation in PM10D-Exposed Mice

As shown in Figure 1, the major phytochemical from LM was identified by UPLC-QTof MS analysis. The details for the identification were as follows. The peak had an [M–H]− at *m*/*z* 739.2084 and fragment ions at *m*/*z* 285 for the aglycon kaempferol (loss of 454 amu for di-rhamnosyl-galactoside). On the basis of this information, the peak was assigned to kaempferol-3-O-(2,6-di-O-α-rhamnopyranosyl-β-galactopyranoside), a well-known compound in *L. mauritiana*, and was tentatively identified as mauritianin (MT).

The effect of LM on airway inflammation was investigated in PM10D-exposed mice. Body weight change and food intake were not significantly different among groups. All mice survived and there were no other abnormal symptoms during the experiment. Airway neutrophilia, a common feature of chronic respiratory inflammation disease, is associated with disease progression [15]. Figure 2B–D shows the number of neutrophils in the BALF and the total numbers of BALF cells and lung cells after administering the LM extract in an animal model of respiratory damage. The exposure of the mice to PM10D for 12 days increased the number of neutrophils in the BALF, and this neutrophil infiltration was limited following the administration of LM extract (Figure 2B). Similarly, the total numbers of BALF and lung cells increased following exposure to PM10D, and the increased cell numbers decreased in LM extract–treated mice (Figure 2C,D).

### 3.2. Effects of LM Extract on the Number of White Blood Cells

As immune cells such as CD4 T cells migrate to lung tissue due to the lung inflammation caused by PM10D, the hematological analysis of white blood cells showed a reduced cell number following PM10D exposure. However, these levels recovered following administration of dexamethasone and LM extract (Figure 3A). WBC differential counting results indicate the percentages of each type of leukocytes (neutrophils, lymphocytes, monocytes, eosinophils, and basophils) that are present in the blood. The number of neutrophils in particular increased in PM10D-exposed mice and significantly decreased following the administration of dexamethasone or LM extract (Figure 3B). These results are consistent with an increase in neutrophils in BALF as seen in Figure 1B. Furthermore, the decreased number of lymphocytes in PM10D-exposed mice recovered following LM administration (Figure 3B). The levels of other white blood cells (i.e., eosinophils, monocytes, and basophils) did not change (Figure 3C).

### 3.3. Effects of LM Extract on the Secretion of Inflammatory Mediators in BALF

The secretion of mediators containing cytokines and chemokines contributes to the pathology of airway inflammatory [16]. Figure 4A–E shows that IL-1α, IL-17, CXCL1, TNF-α, and MIP-2 levels in the BALF were elevated by PM10D exposure and then significantly inhibited by the administration of dexamethasone or LM extract. This effect was dose-dependent and the high dose of LM exerted almost the same effect as dexamethasone.

### 3.4. Effect of LM Extract on Lung Histopathology

To investigate the activities of LM extract in the histopathological analysis of PM10D-induced airway inflammation, the lungs were stained using Masson’s trichrome or H and E. Thickening of the airway wall, inflammatory cell infiltration around the airway, and collagen fibrosis were observed in the lung sections of the PM10D-treated group, and this airway inflammation was reduced in the mice treated with dexamethasone or LM extract (Figure 5A,B). These results indicate that LM extract prevented the histopathological changes in airway inflammation in the lungs of PM10D-treated mice.

### 3.5. Effects of LM Extract on the Expression of Inflammatory Mediators in the Lungs

To examine the effects of LM extract on airway inflammation, the mRNA expression levels of inflammation-related genes were investigated in the lung tissue. Figure 6 shows that mRNA expression of CXCL1, transient receptor potential (TRP) vanilloid 1 (TRPV1), TRP ankyrin 1 (TRPA1), MIP-2, TNF-α, and mucin 5AC (MUC5AC) was elevated in the lung tissues from the group treated only with PM10D compared with the standard (healthy normal control) group and was significantly suppressed by the administration of dexamethasone or LM extract. This effect on the mRNA expression was dose-dependent and the high dose of LM exerted almost the same effect as dexamethasone.

### 3.6. Effects of LM Extract on Immune Cell Numbers in the BALF and Lungs

The effect of LM extract on the change in immune cell numbers was investigated through flow cytometry analysis of the BALF and lungs. The numbers of neutrophils in the BALF and lungs were elevated following exposure to PM10D and were reduced following the administration of LM extract (50 or 100 mg/kg) (Table 2). The absolute number of CD4+, CD8+, and CD62L−/CD44high+-activated T lymphocytes and Gr-1+SiglecF− cells in the BALF increased following PM10D exposure and decreased following the administration of dexamethasone or LM extract (50 or 100 mg/kg). In addition, the absolute numbers of CD62L−/CD44high+ T cells, Gr-1+SiglecF− neutrophils, CD21+/CD35+ B220+ B cells, and Gr-1+CD11b+ myeloid cells in the lungs were elevated by PM10D exposure and decreased following the administration of dexamethasone or LM extract (50 or 100 mg/kg). These results indicate that LM extract ameliorated the neutrophil-dependent airway inflammation caused by exposure to PM10D.

### 3.7. Effects of LM Extract on the MAPK/NF-κB Pathway

To identify the potential pathways responsible for regulating the inhibitory effects of LM extract on the airway inflammatory responses in PM10D-induced mice, we analyzed the MAPK/NF-κB pathway in the lungs (Figure 7, Appendix A). Dexamethasone as a positive control is a glucocorticoid that is available and is used for the treatment of various inflammatory conditions. The phosphorylation of p38, ERK, and JNK was elevated by PM10D exposure and suppressed to the normal (negative control) level by dexamethasone or LM extract (100 mg/kg) administration. The low dose of LM decreased the p-p38 level significantly, but this effect was less potent than that observed by the high dose. This effect on the phosphorylation of p38, ERK, and JNK was dose-dependent and the high dose of LM exerted almost the same effect as dexamethasone. NF-κB–p65 phosphorylation was also increased by PM10D exposure and decreased to the normal (negative control) level following the administration of dexamethasone or LM extract (50 and 100 mg/kg). The protective effects of LM were similar to those of dexamethasone. These results demonstrate that the inhibitory effects of LM extract on airway inflammation are due to the suppression of phosphorylation of p38, ERK, JNK, and NF-κB in the MAPK/NF-κB signaling pathway.

### 3.8. Expectorant Effect of LM Extract According to Phenol Red Secretion

To evaluate the effect of LM extract on expectoration, the sputum excretion content in the trachea was measured using the phenol red secretion method in ICR mice. Since phenol red is a carcinogen that can cause toxicity, the ICR mouse model, a model commonly used in toxicity studies, was used. Oral administration of 200 mg of Levosol/kg (the positive control) and 100 mg of LM extract/kg significantly increased phenol red release compared with the control (1.37-fold and 1.57-fold, respectively) (Figure 8). These results illustrate the expectorant activity of LM extract.

## 4. Discussion

Airway inflammation, characterized by the mobilization and activation of immune cells (mainly neutrophils) and the immoderate production of molecular mediators, is considered a primary component of the pathogenesis of lung parenchymal destruction (emphysema) and airway remodeling (chronic bronchitis) in respiratory disorders such as chronic obstructive pulmonary disease (COPD) [17]. Neutrophils must be inhibited in inflamed airways to allow individuals to recover from lung damage in chronic respiratory disease [18].

Some pathological features similar to COPD were reproduced in PM10D-exposed experimental mice, including the agglutination and penetration of immune cells, the stenosis of the small airway, airway wall thickening, and pulmonary collagen fibrosis [19,20,21]. However, the histopathological changes in the lungs improved in the groups that received LM extract compared with the PM10D control group. The differential immune cell counts demonstrated that the BALF of normal mice had few neutrophils, eosinophils, and lymphocytes and that PM10D exposure obviously changed this cell composition. The numbers of eosinophils, lymphocytes, and, predominantly, neutrophils were increased in PM10D-exposed control mice. Furthermore, the infiltration and persistent recruitment of neutrophils were induced in the lung tissues, a typical feature of COPD. Compared with the PM10D controls, the elevated number of neutrophils and the increased expression of inflammatory cytokines (such as IL-1α, IL-17, MIP-2, CXCL1, and TNF-α) in the lungs and BALF were obviously lower in the LM groups, markedly reducing neutrophilic inflammation.

Previous studies have indicated that PM-induced respiratory inflammation mainly relates to small airway disease, lung parenchyma, and chronic bronchitis and is characterized by the recruitment and persistence of immune cells, predominantly neutrophils [22,23]. Neutrophils contribute to airway remodeling by producing proinflammatory cytokines containing TNF-α and IL-1 [24]. The functions of neutrophils and proinflammatory mediators in the pathophysiology of airway inflammatory disease are well known. TNF-α is a major neutrophil chemoattractant and multifunctional proinflammatory cytokine [25]. IL-1α stimulates the migration of neutrophils by inducing CXCL1, a key chemoattractant of neutrophils [26]. Studies have shown that IL-17(A) and TNF-α increase endothelial expression of the neutrophilic chemokines CXCL2 (MIP-2), CXCL1, and CXCL5 and enhance the influx of neutrophils to inflammation sites [27,28]. In this study, the increased expression levels of TRPA1, TRPV1, and MUC5AC in the lungs of PM10D-exposed control mice were inhibited by the administration of LM extract. The TRP receptor proteins (e.g., TRPV1 and TRPA1), which are irritant-sensing ion channels expressed in airway chemosensory nerves, are related to leukocyte infiltration in the airway, cytokine production, and cough response [29,30]. MUC5AC mucin is a main constituent of airway mucus, and mucus hypersecretion is a pathological feature of airway diseases [31]. Thus, the decrease in TRPA1, TRPV1, and MUC5AC in the lungs following the administration of LM extract should improve respiratory disease symptoms, such as cough and mucus production, in mice.

In the present study, we determined the mechanism responsible for these effects of LM. We investigated the activation of the MAPK/NF-κB signaling pathway in the lungs of PM10D-exposed mice and found that the increased ERK/p-38/JNK MAPK and NF-κB signals were suppressed by the administration of LM extract. These pathways are responsible for regulating the production of inflammatory mediators and the consequential neutrophilic infiltration and activation in chronic pulmonary disease [32]. Therefore, our results suggest that the reduction in neutrophilic airway inflammation following the administration of LM extract may be majorly attributed to the suppression of the MAPK/NF-κB signaling pathway.

According to the UPLC-QTof MS analysis, kaempferol 3-O-(2,6-di-O-α-rhamnopyranosyl)-β-galactopyranoside, also termed mauritianin, was the main flavonoid glycoside found in LM. From the whole plant of LM, mauritianin was isolated together with kaempferol-3-O-α-rhamnopyranosyl-(1–2)-β-galactopyranoside, kaempferol-3-O-robinobioside, and hyperin [7]. Mauritianin is a flavonoid that is rare in plants. As previously reported, mauritianin showed cytoprotective, antioxidant, neuroprotective, and antitumor-promoting activities [33,34,35]. However, mauritianin has not yet been reported to have effects against airway inflammation or related pulmonary diseases. Previous phytochemical investigations showed that *Lysimachia* species contain triterpenoid saponins and flavonoids [36]. Saponins in plants are important natural anti-inflammatory compounds that act on the activity of several proinflammatory cytokines in various inflammatory models [37]. These findings suggest that these compounds from LM could be potential anti-inflammatory agents for the protection of respiratory diseases. To our knowledge, this is the first study to demonstrate that LM exhibits an anti-inflammatory effect against inflammation-induced respiratory disease. Further investigation into the effects of other compounds with mauritianin from LM on airway inflammation is therefore required.

## 5. Conclusions

LM extract ameliorated neutrophilic airway inflammation via the suppression of the MAPK/NF-κB signaling pathway in mice with PM10D-induced respiratory inflammation. The results of our study suggest that LM may be an effective candidate for the prevention/treatment of respiratory disease.

## Figures and Tables

**Figure 1 nutrients-16-03732-f001:**
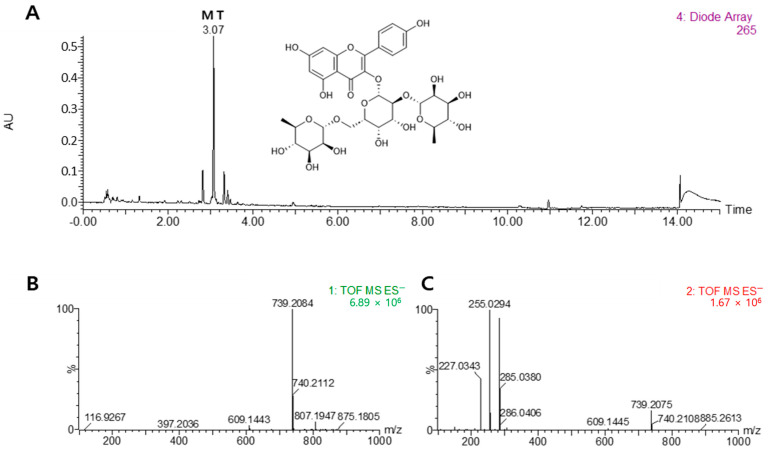
Chemical structure of mauritianin (MT). Comparison of UPLC DAD chromatograms of the 50% ethanol extract from *Lysimachia mauritiana* (**A**), MS (**B**), and MS/MS (**C**) data for the qualitative analysis of major chemical constituents. The UPLC chromatogram was acquired at 265 nm.

**Figure 2 nutrients-16-03732-f002:**
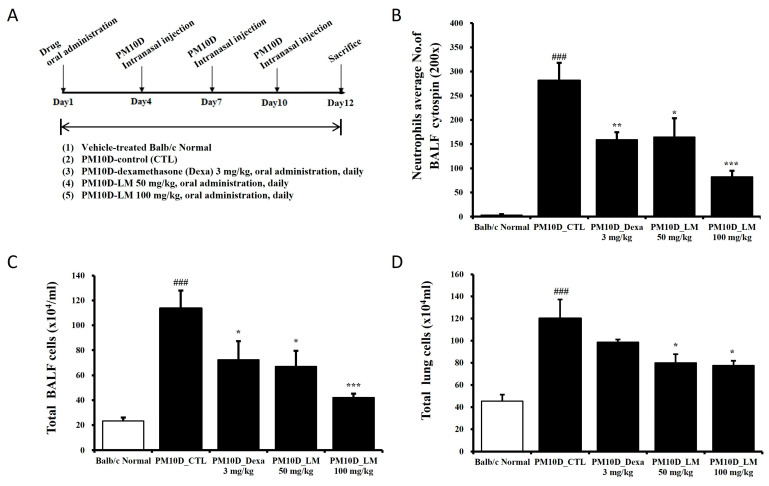
Experimental setup and the effect of *Lysimachia mauritiana* extract on total and immune cell numbers in a model of airway inflammation induced by particulate matter with a diameter less than 10 µm plus diesel exhaust particles (PM10D) [13]. (**A**) Experimental setup; (**B**) neutrophils in bronchoalveolar lavage fluid (BALF) cytospin (magnification: 200×); total numbers of (**C**) BALF cells and (**D**) lung cells. N = 6/group. ### *p* < 0.001 vs. normal. * *p* < 0.05, ** *p* < 0.01, and *** *p* < 0.001 vs. PM10D control (CTL).

**Figure 3 nutrients-16-03732-f003:**
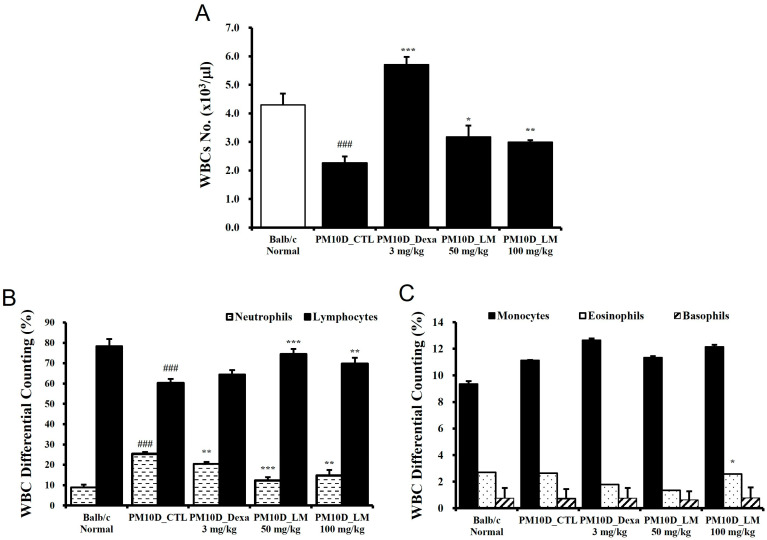
Effect of *Lysimachia mauritiana* extract on (**A**) the number of white blood cells (WBCs) and (**B**,**C**) WBC differential cell counting. N = 6/group. ### *p* < 0.001 vs. normal. * *p* < 0.05, ** *p* < 0.01 and *** *p* < 0.001 vs. particulate matter with a diameter less than 10 µm plus diesel exhaust particles (PM10D) control (CTL).

**Figure 4 nutrients-16-03732-f004:**
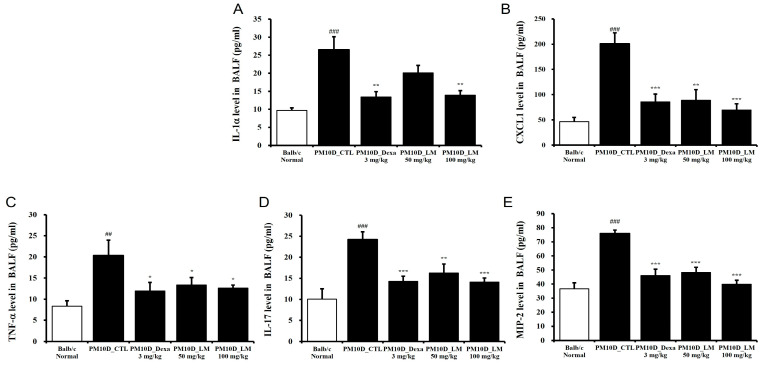
Effect of *Lysimachia mauritiana* extract on the release of cytokines and chemokines in bronchoalveolar lavage fluid (BALF) in a model of airway inflammation induced by particulate matter with a diameter less than 10 µm plus diesel exhaust particles (PM10D). BALF production of (**A**) IL-1α, (**B**) CXCL1, (**C**) TNF-α, (**D**) IL-17, and (**E**) MIP-2 (n = 6/group). ## *p* < 0.01 and ### *p* < 0.001 vs. normal. * *p* < 0.05, ** *p* < 0.01, and *** *p* < 0.001 vs. PM10D control (CTL).

**Figure 5 nutrients-16-03732-f005:**
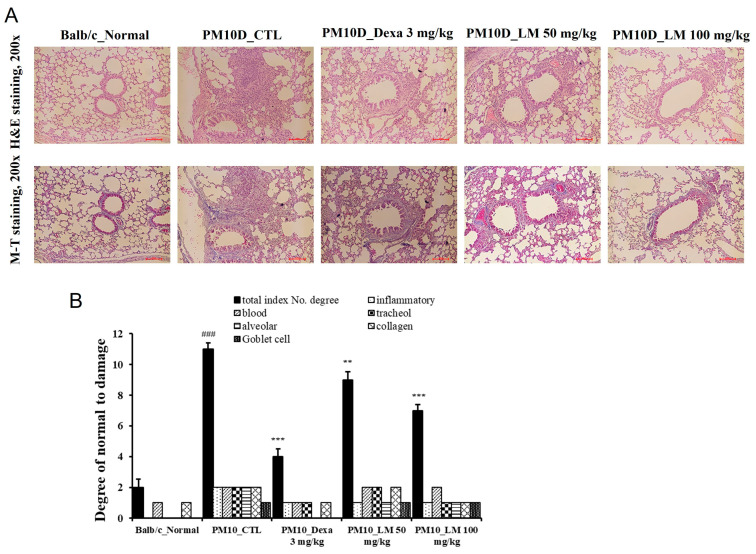
Effect of *Lysimachia mauritiana* extract on lung histopathology. (**A**) Hematoxylin and eosin (H and E) staining and Masson’s trichrome (MT) staining of the lung tissue of mice with airway inflammation induced by particulate matter with a diameter less than 10 µm plus diesel exhaust particles (PM10D) (magnification: 200×) (n = 6). (**B**) Histopathological cell damage. ### *p* < 0.001 vs. normal., ** *p* < 0.01 and *** *p* < 0.001 vs. PM10D control (CTL).

**Figure 6 nutrients-16-03732-f006:**
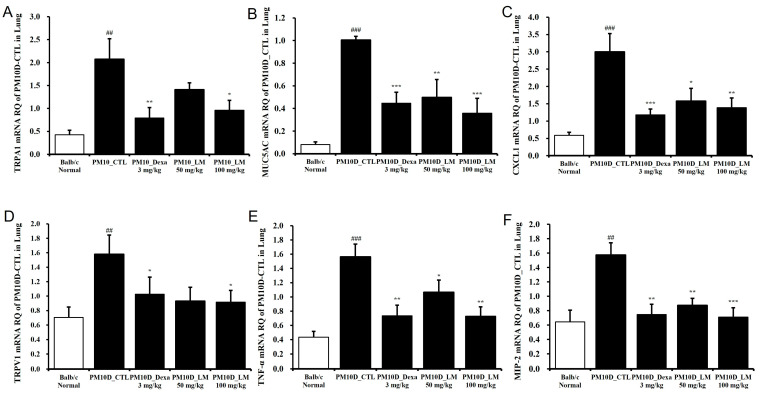
Effect of *Lysimachia mauritiana* extract on the mRNA expression of airway inflammation–related genes in the lung tissue of mice with airway inflammation induced by particulate matter with a diameter less than 10 µm plus diesel exhaust particles (PM10D). mRNA expression levels of (**A**) TRPA1, (**B**) MUC5AC, (**C**) CXCL1, (**D**) TRPV1, (**E**) TNF-α, and (**F**) MIP-2 (n = 6). ## *p* < 0.01 and ### *p* < 0.001 vs. normal. * *p* < 0.05, ** *p* < 0.01, and *** *p* < 0.001 vs. PM10D control (CTL).

**Figure 7 nutrients-16-03732-f007:**
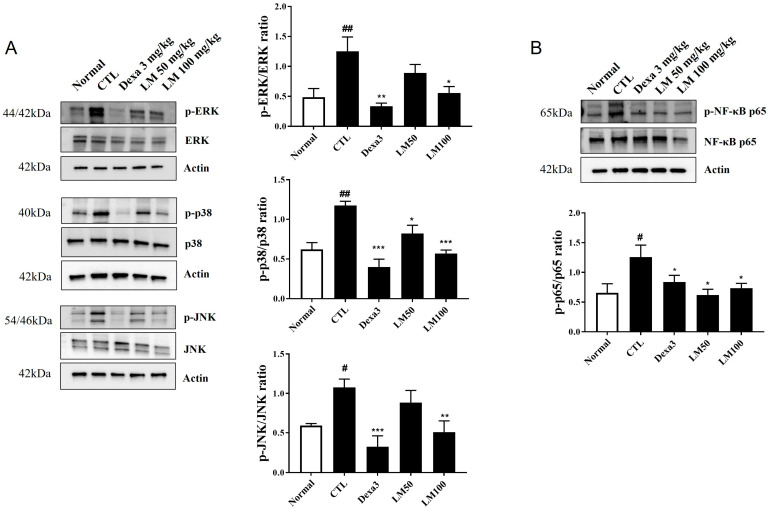
Effect of *Lysimachia mauritiana* extract on mitogen-activated protein kinase (MAPK)/nuclear factor-kappa B (NF-κB) signaling induced by particulate matter with a diameter less than 10 µm plus diesel exhaust particles (PM10D) in the lung tissue of mice with PM10D-induced airway inflammation. (**A**) Protein expression of phospho-ERK, ERK, phospho-p38, p38, phospho-JNK, JNK, phospho-p65, p65, and β-actin. (**B**) Quantitative analysis of each protein band using ImageJ (n = 3). # *p* < 0.05 and ## *p* < 0.01vs. normal. * *p* < 0.05, ** *p* < 0.01, and *** *p* < 0.001 vs. PM10D control (CTL).

**Figure 8 nutrients-16-03732-f008:**
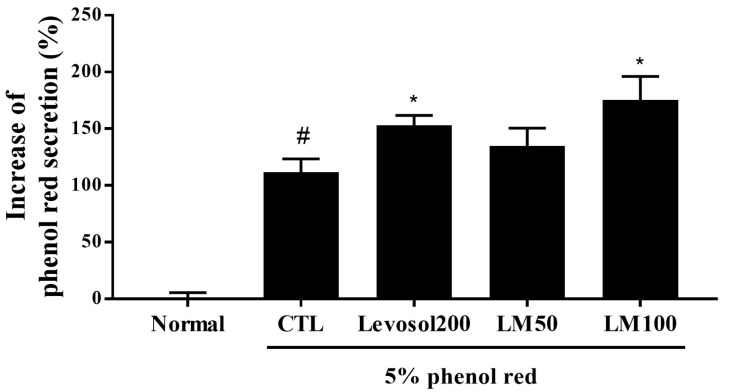
Effect of *Lysimachia mauritiana* extract on phenol red secretion in ICR mice. The amount of phenol red secretion in the airways was measured by injecting 5% phenol red into mice treated with Levosol (positive control) or LM extract for 3 days. N = 12/group. # *p* < 0.05 vs. normal, and * *p* < 0.05 vs. control.

**Table 1 nutrients-16-03732-t001:** Primers’ sequence used in qRT-PCR analysis.

Gene	Primer	Oligonucleotide Sequence (5′-3′)
*Actin*	F	TGGAATCCTGTGGCATCCAT
	R	TAAAACGCAGCTCGTAACAG
*TNF-* *α*	F	CCTGTAGCCCACGTCGTAGC
	R	TTGACCTCAGCGCTGAGTTG
*MIP-2*	F	ATGCCTGAAGACCCTGCCAAG
	R	GGTCAGTTAGCCTTGCCTTTG
*CXCL1*	F	CCGAAGTCATAGCCACAC
	R	GTGCCATCAGAGCAGTCT
*MUC5AC*	F	AGAATATCTTTCAGGACCCCT
	R	ACACCAGTGCTGAGCATACTT
*TRPV1*	F	CATCTTCACCACGGCTGCTTAC
	R	CAGACAGGATCTCTCCAGTGAC
*TRPA1*	F	TGAGATCGACCGGAGT
	R	TGCTGAAGGCATCTTG

Abbreviations: F, forward; R, reverse.

**Table 2 nutrients-16-03732-t002:** The effects of LM extract on airway immune cell numbers and neutrophilic airway inflammation in PM10D-induced inflammation model; fluorescence-activated cell sorting (FACS) analysis.

Cell Types	Absolute No. (Mean ± Standard Error of the Mean)				
	Balb/c Normal	PM10D-CTL	PM10D-Dexa 3 mg/kg	PM10D-LM 50 mg/kg	PM10D-LM 100 mg/kg
BALF					
Lymphocytes (×10^4^ cells)	2.73 ± 0.68	7.45 ± 1.76 ^#^	4.20 ± 1.02	4.46 ± 0.35	3.06 ± 0.26 *
Neutrophils (×10^4^ cells)	5.41 ± 1.14	53.09 ± 6.44 ^###^	21.19 ± 5.56 **	26.32 ± 6.81 **	9.98 ± 2.49 ***
Eosinophils (×10^4^ cells)	11.74 ± 3.89	48.90 ± 12.93 ^##^	44.74 ± 16.68	34.26 ± 10.56	27.26 ± 3.56
CD4^+^ (×10^4^ cells)	0.55 ± 0.25	31.56 ± 8.11 ^##^	10.51 ± 2.32 **	14.63 ± 3.60	6.85 ± 0.61 **
CD8^+^ (×10^4^ cells)	0.10 ± 0.06	17.01 ± 2.15 ^###^	3.98 ± 1.01 ***	6.92 ± 1.47 **	5.23 ± 2.68 **
CD62L^−^/CD44^high+^(×10^4^cells)	1.64 ± 0.39	89.75 ± 15.92 ^###^	55.28 ± 15.35	44.74 ± 11.58	25.01 ± 2.27 ***
Gr-1^+^SiglecF^−^ (× 10^4^ cells)	1.09 ± 0.53	53.18 ± 9.61 ^###^	14.35 ± 4.32 **	25.72 ± 6.35 *	6.91 ± 2.42 ***
Lung					
Lymphocytes (× 10^4^ cells)	14.46 ± 1.96	24.06 ± 4.26 ^#^	33.31 ± 2.46	21.02 ± 4.11	21.75 ± 0.65
Neutrophils (× 10^4^ cells)	24.15 ± 5.70	81.34 ± 17.00 ^##^	50.86 ± 2.19	47.85 ± 6.31	42.20 ± 5.42 *
Eosinophils (× 10^4^ cells)	5.96 ± 0.73	12.20 ± 2.48 ^#^	12.53 ± 0.55	9.37 ± 0.37	12.10 ± 0.99
CD4^+^ (× 10^4^ cells)	15.94 ± 3.51	34.79 ± 5.47 ^##^	33.93 ± 1.28	31.43 ± 4.08	31.49 ± 2.09
CD8^+^ (× 10^4^ cells)	6.68 ± 1.49	21.77 ± 4.24 ^##^	17.45 ± 0.79	13.71 ± 2.12	12.80 ± 1.41 *
CD4^+^CD69^+^ (× 10^4^ cells)	1.38 ± 0.44	4.06 ± 0.67 ^##^	2.41 ± 0.34 *	3.54 ± 0.47	2.56 ± 0.42
CD62L^−^/CD44^high+^(× 10^4^ cells)	3.88 ± 0.76	17.88 ± 1.47 ^###^	9.00 ± 1.01 ***	11.62 ± 3.24	10.03 ± 2.06 **
CD21^+^/CD35^+^B220^+^ (× 10^4^ cells)	4.33 ± 2.15	21.08 ± 4.84 ^##^	8.09 ± 2.14 *	9.99 ± 1.05 *	6.59 ± 0.16 **
Gr-1^+^SiglecF^−^ (× 10^4^ cells)	9.49 ± 2.76	41.83 ± 3.15 ^###^	19.26 ± 2.26 ***	25.51 ± 2.61 **	17.64 ± 3.23 ***
Gr-1^+^CD11b^+^ (×10^4^ cells)	15.58 ± 3.13	56.92 ± 7.92 ^###^	28.78 ± 2.15 **	33.84 ± 5.44 *	25.32 ± 4.35 **

Abbreviations: CTL, control; Dexa, dexamethasone. The data are presented as means ± SEM (n = 8). ^#^ *p* < 0.05, ^##^ *p* < 0.01, ^###^ *p* < 0.001 versus the Normal group, and * *p* < 0.05, ** *p* < 0.01, *** *p* < 0.001 versus the CTL group, as determined by analyses of variance (ANOVA) followed by Duncan’s multiple range tests.

## Data Availability

The data used to support the findings of this study are included within this article.

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
