# Peer review of "Lysimachia mauritiana Lam. Extract Alleviates Airway Inflammation Induced by Particulate Matter Plus Diesel Exhaust Particles in Mice"

_nutrients, 2024, doi:10.3390/nu16213732_

Round 1
Reviewer 1 Report
Comments and Suggestions for Authors
Dear Authors,
I have read the manuscript and I send you my comments:
1) Please add more data on L mauritiana, explaining how you have choose the dosage
2) results: in figure you show that corticosteroid + L mautiana have the same effects of L mauritiana at higher dosage, please explain why
Comments on the Quality of English Language
none
Reviewer 2 Report
Comments and Suggestions for Authors
The article refers to the curative potential of an ethanolic extract from Lysimachia mauritiana (LM) against airway inflammation caused by atmospheric pollutants, specifically by fine particles (PM10D) emitted in the diesel exhaust air.
Overall, the article is original and interesting, at the same time it covers a topic of great relevance; the problem of pollutants with low molecular mass and augmented inflammatory effect and high allergen potential for humans. The methodology is appropriate; the control negative and the positive control are both present. Also, conclusions are consistent with the results of the study. Tables and Figures are clear and easy to understand.
All the same, there are a number of shortcomings.
-The introduction is insufficient: the literature data on the chemical composition of the vegetal species in general, and the alcoholic extracts from LM in particular, both are absolutely necessary.
-Also, a connection must be made between the chemical composition and the pharmacological activity studied - this connection will establish the probable chemical basis of the ability of LM extract to act against the inflammation process of the respiratory system.
-A minimum of chemical characterization has to be done for the 50% ethanolic extract from Lysimachia mauritiana. The plant species have a different chemical composition during the vegetation period, and from one geographical region to another; thus, the pharmacological results obtained in this study can only be reproduced using an extract which presents a very similar chemical composition; the minimal chemical characterization for a vegetal extract comprises the analyses of total phenolics, of total flavones and of UV-VIS spectrum to reveal the major compounds and peaks which define the working active extract.
-In Figure 1 (B, C) is a missing letter - F from BALF (0 - y axis).
-The results in Figure 2 need to be better detailed.
-The results in Figure 6 need further discussion: the possible side effects of LM administration by decreasing of the mitogen-activated protein kinase (MAPK)/nuclear factor-kappa B (NF-κB) signaling pathway below the negative control series; the comparison of Dexa3’ effects face to LM extract, etc.
Altogether, the article needs a major revision.
Reviewer 3 Report
Comments and Suggestions for Authors
The researchers have evaluated the potential of Lysimachia mauritiana (LM) extract against airway inflammation in a mouse model exposed to a fine dust mixture of diesel exhaust particles and particulate matter with a diameter of less than 10 µm (PM10D). After twelve days, the immune cell subtypes, histopathology, and expression of inflammatory mediators were analyzed from the bronchoalveolar lavage fluid (BALF) and lungs. LM alleviated the accumulation of neutrophils and the number of inflammatory cells in the lungs and BALF of the PM10D-exposed mice. LM also reduced the release of inflammatory mediators (MIP-2, IL-17, IL-1α, CXCL1, TNF-α, MUC5AC, and TRP receptor channels) in the BALF and lungs. Lung histopathology was used to examine airway inflammation and accumulation of collagen fibres and inflammatory cells after PM10D exposure, and it was shown that LM administration ameliorated this inflammation. Furthermore, LM extract inhibited the lung's MAPK and NF-κB signalling pathway and improved expectoration activity through an increase in phenol red release from the trachea.
This detailed and robust study demonstrated the protective effects of an extract of Lysimachia mauritiana on PM10D-triggered airway inflammation in mice. The general findings are like those the research group made with an extract of Siraitia grosvenorii (Sung et al., 2023). However, in both cases, the composition of bioactive factors in the extracts has not been analyzed. A vague indication of some bioactive factors that should be there is given in the text but not specifically evaluated in the tested product. If the present study is to rise above the level of 'another ill-defined plant extract with beneficial properties on health', the authors need to give a more detailed description of the bioactive factors/metabolites present in the extract and speculate how these factors may modulate their protective effect.
Title: ‘Lysimachia mauritiana Lam. Extract Alleviates Airway Inflammation’. Mice are dosed with LM for three days before exposure to PM10D, so the observed protective actions may be due to adaptational changes [see Figure 7] that limit or prevent the direct action of PM10D rather than alleviating its deleterious inflammatory actions. While the present finding is interesting, there is no guarantee that similar beneficial responses to LM would be evident if the mice already had established PM10D-induced inflammation.
Ln 61-64 Give more details about the source material, the extraction procedure and reasons for choosing it, and the bioactive factors/metabolites present in the extract.
Ln 65 Were the mice and housing facilities specific-pathogen-free (SPF)?
Ln 74 How were doses established? Are they physiological or supra-physiological?
Ln 77 Anaesthesia or form of euthanasia?
Ln 78 Why was a different strain of mice used for experiment 2?
Ln 91 Published method?
Ln 132 Basic information on mouse body weight change, food intake, health status, and survival data should be provided since this is the starting point for all subsequent analyses.
Ln 139 ‘decreased following the administration of LM extract (Fig. 1B).’ See general comments. Mice treated with LM before exposure to PM10D, so subsequent beneficial responses may be prophylactic rather than treatment effects.
Ln 177 ‘(Fig. 4A–C). These results indicate that LM extract recuperated the histopathological’. No graph C is in Figure 7. Also, 'recuperated' is not the proper word here since LM was given to mice before PM10D-induced inflammation.
Ln 233 Why was a different strain of mice used here?
Ln 189 ‘tissues of the control group with respiratory damage induced by PM10D compared with the normal group’. Clarify. Tissues from the group treated only with PM10D compared with the standard (healthy control) group.
Sung et al. (2023). Siraitia grosvenorii extract attenuates airway inflammation in a mouse model of respiratory disease induced by particulate matter 10 plus diesel exhaust particle. Nutrients 2023, 15, 4140. https://doi.org/10.3390/nu15194140
Round 2
Reviewer 2 Report
Comments and Suggestions for Authors
The authors completed the studies with the requested information. Thus:the compounds potentially responsible for the pharmacological effects were indicated; the vegetal extract was characterized in terms of key active compounds, so it can be reproduced; the study results in Figures were better explained; the discussion section was completed for better understanding the idea and the results of the studies.The material and method section has been completed with in vivo study design data.
Reviewer 3 Report
Comments and Suggestions for Authors
All issues raised during review have been dealt with in a thorough and robust manner.